

# Kinetic and thermodynamic analyses of the corrosion inhibition of synthetic extracellular polymeric substances

Liew Chien Go[1], Dilip Depan[1], William E. Holmes[2], August Gallo[3], Kathleen Knierim[3], Tre Bertrand[4] and Rafael Hernandez[1,2]

[1] Department of Chemical Engineering, University of Louisiana at Lafayette, Lafayette, LA, USA
[2] Energy Institute of Louisiana, University of Louisiana at Lafayette, Lafayette, LA, USA
[3] Department of Chemistry, University of Louisiana at Lafayette, Lafayette, LA, USA
[4] Coastal Chemical Co. LLC, Broussard, LA, USA

## ABSTRACT

**Background:** Extracellular polymeric substances (EPS) extracted from waste activated sludge (WAS) have previously shown its potential in corrosion inhibition. The aim of this study is to design a synthetic EPS formulation as a surrogate of natural WAS EPS to overcome the corrosion inhibition inconsistency in WAS EPS. The adsorption behavior of the designed inhibitor was studied by kinetic and thermodynamic analyses.

**Methods:** Synthetic EPS is a bio-inspired material that was formulated based on the most typical chemical compositions of natural WAS EPS, that is, proteins, carbohydrates, humic substances, nucleic acids, and uronic acids, which was not optimized for corrosion inhibition performance. It is a mixture of glutamic acid, carboxymethylcellulose, humic acid, thymine, and alginic acid. Its corrosion inhibition performance was tested with carbon steel in 3.64% NaCl saturated with $CO_2$, using the potentiodynamic polarization scanning technique. The resulted electrochemical parameters were used to evaluate the empirical corrosion kinetic and thermodynamic adsorption parameters.

**Results:** Addition of synthetic EPS showed significant decrease in corrosion rate as compared to the control. The inhibition efficiency improved with increasing inhibitor concentration and temperature. The optimum performance was 94% with 204 mg/L of inhibitor applied at 70 °C (343 K). The inhibition performance was controlled by both the concentration of inhibitor and temperature. Chemisorption of the inhibitor molecules contributed to the overall inhibition performance, reducing the contact of metal with the corrosive environment, thus, slowing down the overall corrosion rate.

## INTRODUCTION

Extracellular polymeric substances (EPS) are the metabolic products produced by most microorganisms. They accumulate on the surface of microorganisms, acting as protective barriers against the microorganisms' external environment (*Liu & Fang, 2002*). Typically,

Corresponding author
Rafael Hernandez,
rafael.hernandez@louisiana.edu

carbohydrates have been identified as the major constituents in the EPS of many pure cultures (*Cescutti et al., 1999*; *Kennedy & Sutherland, 1996*), whereas proteins were found in substantial quantities in the sludge of many wastewater treatment reactors (*Liu & Fang, 2002*; *Veiga et al., 1997*). Small amounts of humic substances (*Nielsen, Griebe & Nielsen, 1995*), uronic acids, and nucleic acids (*Liu & Fang, 2002*; *Frølund et al., 1996*) were also detected in EPS. A previous study (*Go et al., 2019*; *Go, Holmes & Hernandez, 2019*) showed the potential of EPS extracted from waste activated sludge (WAS) of wastewater treatment operations as a green corrosion inhibitor for $CO_2$ corrosion. A maximum inhibition performance of about 80% was achieved with the application of 1,000 mg/L of this inhibitor. The corrosion inhibition mechanism of WAS EPS was explained by the formation of a biofilm on the metal surface, shielding the metal surface from the corrosive environment. Even though the inhibition performance is comparable to commercial products, the nature of WAS caused inconsistency in inhibition efficiency. The composition of WAS is dependent on wastewater treatment operational parameters, such as inlet biochemical oxygen demand and sludge residence time.

The present study focused on the evaluation of a corrosion inhibitor from the surrogate of WAS EPS. The reason of making the surrogate was to have control on the chemical composition of the corrosion inhibitor used and ensure consistent inhibition performance. This study hypothesized that the designed synthetic EPS will demonstrate similar corrosion inhibition behavior as the natural WAS EPS because it was formulated based on the major chemical compositions of natural WAS EPS. The novelty of this research was the design of a surrogate biomass-based corrosion inhibitor inspired by sources with varied chemical compositions. To the knowledge of the authors, this line of work has not been reported elsewhere. This study is unique in the way that it is a multidisciplinary work. Bio-inspired systems and materials are not uncommon in the literature. Yet, this concept is the pioneer of the field of corrosion inhibitor formulation development. For instance, some of the most commonly applied programing algorithms in computer science and engineering today are bio-inspired. Algorithms like genetic and ant colony mimic the natural biological systems to solve research problems. This study is adopting the bio-inspired concept into the corrosion inhibitor formulation development. It is believed that this line of multidisciplinary work could benefit and advance the research in corrosion inhibitor development, especially the renewable type.

The present study seeks to investigate the corrosion inhibitive properties of synthetic EPS for carbon steel in 3.64% NaCl solution saturated with $CO_2$ gas using the potentiodynamic polarization technique. The corrosion kinetic parameters and thermodynamic adsorption parameters are calculated and reported.

## MATERIALS AND METHODS

### Metal specimen preparation

Potentiodynamic polarization scans were performed on carbon steels of the following weight percentage composition: 0.17 C, 0.08 Mn, 0.014 P, 0.002 S, 0.022 Si, 0.02 Cu, 0.01 Ni, 0.04 Cr, 0.002 Sn, 0.042 Al, 0.006 N, 0.001 V, 0.0001 B, 0.001 Ti, 0.001 Cb, and the remainder iron. The pre-treatment of the specimens' surface was carried out by grinding

**Table 1 Details of synthetic EPS.**

| Type | Chemical | Vendor | Specification | Amount (mg/L) |
|------|----------|--------|---------------|---------------|
| Protein | Glutamic acid | Acros Organics | 99% | 30 |
| Carbohydrate | Carboxymethyl cellulose (CMC) | Acros Organics | MW 70,000 | 12 |
| Humic acid | Humic acid | Acros Organics | 45–70% | 5 |
| Nucleic acid | Thymine | Alfa Aesar | 97% | 2 |
| Uronic acid | Alginic acid, sodium salt | Acros Organics | – | 2 |

with sandpapers of 40, 220, 320 grits, rinsing with deionized water, then drying with paper towel. The specimens were used immediately after pre-treatment.

## Corrosive medium preparation

The corrosive medium was prepared with 36.4 g of NaCl (Fisher Scientific, Hampton, NH, USA) in one L of deionized water to make up 3.64% of NaCl solution. The deionized water used was drinking water filtered with Milli-Di Water Purification System (Merck Millipore, Burlington, MA, USA). Prior to starting of each experiment, $CO_2$ gas was sparged in the test solution at 30 psi for 30 min. Then, the solution was transferred into the reactor, with $CO_2$ gas continuously sparging throughout the experiment at 20 psi.

## Corrosion inhibitor preparation

A mixture of several chemical compounds was labeled as synthetic EPS. It was used as the test corrosion inhibitor in this study. The details of each compound, that is, chemical type, compound identity, vendor, specification, and composition, are listed in Table 1. These compounds were mixed in the given composition as synthetic EPS. The concentrations of inhibitors used in the following runs were doubled, tripled, and quadrupled.

## Potentiodynamic polarization method

Potentiodynamic polarization experiments were carried out with Gamry Flexcell Critical Pitting Cell Kit, connecting to the Gamry Potentiostat Interface 1000. The reference, counter, and working electrodes used were saturated calomel electrode, graphite rod, and the metal specimen, respectively. The setup was equipped with a heating jacket connected to TDC4 Omega temperature controller to maintain the test solution at a desired temperature, in this case, 25, 50, and 70 °C. The Glas-Col GT Series stirrer was connected to the setup externally and adjusted to 50 rpm to get the desired shear and to ensure even heating. The working solution volume was one L. The working area of the metal specimens had a circular form of 5 cm$^2$.

The potentiodynamic polarization scans were carried out in potential range of −0.25 to +0.25 V vs corrosion potential ($E_{corr}$) at a scan rate of 3 V/hr. Corrosive medium was added into the reactor with carbon dioxide gas sparging constantly at 20 psi throughout the experiment. The reactor was allowed to equalize for 30 min prior to the beginning of experiment. After the system was equalized, Tafel plots were graphed with Gamry DC105 DC Corrosion Technique Software until three relatively similar readings were obtained. Next, corrosion inhibitor was added into the reactor. The reactor was again

allowed to equalize for 30 min, then Tafel plots were graphed. This step was repeated until three consecutive graphs with similar trends were yielded, to ensure the stability of the system. Subsequently, the concentration of the corrosion inhibitor was increased. Again, the system was being equalized for 30 min, followed by the graphing of Tafel plots.

The Tafel plot was plotted with the mean values of corrosion potential ($E_{corr}$) and corrosion current density ($I_{corr}$) from the triplicates of the experiments, while the electrochemical parameters obtained from the curves were reported with mean and standard deviation. The corrosion current densities were found by extrapolating the linear Tafel segment of the anodic and cathodic curves to the corrosion potential. The corrosion inhibition efficiency was then calculated with Eq. (1).

$$\text{Inhibition Efficiency } (\%) = \frac{I_{\text{corr, uninhibited}} - I_{\text{corr, inhibited}}}{I_{\text{corr, uninhibited}}} \times 100\% \qquad (1)$$

## Fourier-transform infrared spectroscopy

Agilent Cary 630 Fourier-transform infrared spectroscopy (FTIR) incorporated with MicroLab software were used for the FTIR analysis in this study. This equipment worked based on attenuated total reflection Method. The scanning was range between 4,000 to 400 $cm^{-1}$ with resolution of 4 $cm^{-1}$.

## RESULTS

### Corrosion inhibition performance

The Tafel plots generated from the potentiodynamic polarization measurements for carbon steel in 3.64% NaCl saturated with $CO_2$ gas with synthetic EPS range from 51 to 204 mg/L at 25, 50, and 70 °C (298 K, 323 K, 343 K) are presented in Figs. 1–3, respectively. The details of electrochemical parameters obtained from the curves, namely corrosion potential ($E_{corr}$), corrosion current density ($I_{corr}$), and inhibition efficiency, are listed in Table 2. Moreover, the effects of inhibitor concentration and media temperature are addressed in the "Discussion" section. It is also worth noting that the significance of operation and economics of the synthetic EPS as an oil field corrosion inhibitor formulation is also included in the "Discussion" section.

### Corrosion kinetic parameters

Corrosion kinetic parameters, that is, apparent activation corrosion energy ($E_a$), enthalpy of activation ($\Delta H_a^\circ$), and entropy of activation ($\Delta S_a^\circ$), are listed in Table 3. Two Arrhenius plots used for the evaluation of these corrosion kinetic parameters are shown in Figs. 4 and 5. The details of calculations are discussed in the "Discussion" section.

### Thermodynamic adsorption parameters

The standard free energy of adsorption ($\Delta G_{ads}^\circ$), enthalpy of adsorption ($\Delta H_{ads}^\circ$), and the entropy of adsorption ($\Delta S_{ads}^\circ$) are listed in Table 4. The Langmuir isotherm and the Van't Hoff plots are shown in Figs. 6 and 7, respectively. The equations and graphs involved for the thermodynamic adsorption parameters are explained in the "Discussion" section.

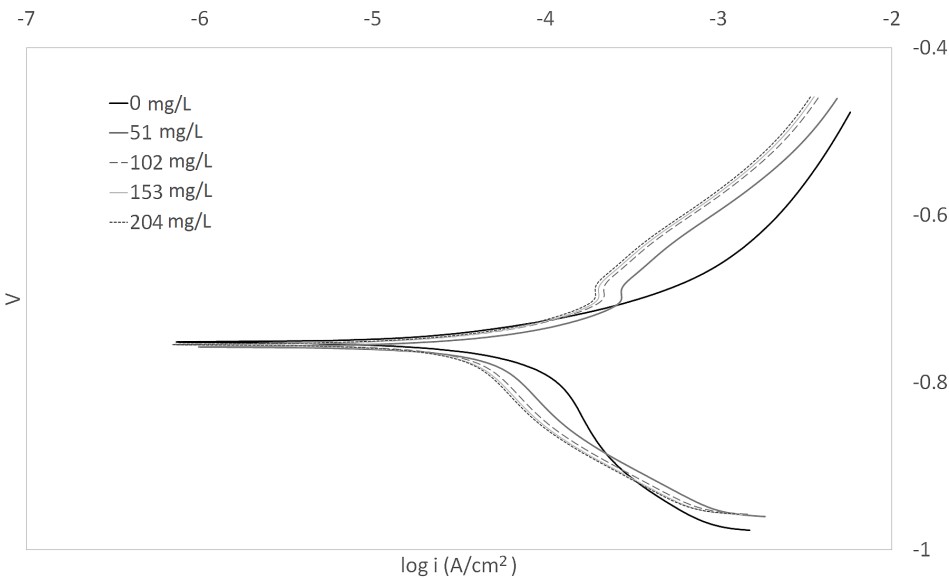

**Figure 1 Tafel plot for carbon steel in 3.64% NaCl concentrated with $CO_2$ with different concentrations of synthetic EPS at 25 °C.**

## Fourier-transform infrared spectroscopy

The IR spectra is shown in Fig. 8 and the characteristic IR absorption frequencies of the responding organic functional groups of synthetic EPS is tabulated in Table 5.

## DISCUSSION

### Properties of synthetic EPS

Synthetic EPS is a mixture of several major groups of chemicals in natural WAS EPS. Although there are many ways to extract EPS and each of the methods give different chemical composition (*Liu & Fang, 2002*; *Frølund et al., 1996*; *Comte, Guibaud & Baudu, 2006*); the composition of synthetic EPS formulated in this study will be based on the method of heating. Typically, the EPS extracted by heating has the highest proteins concentration, followed by carbohydrates, humic substances, nucleic acids, and uronic acids (*Liu & Fang, 2002*; *Frølund et al., 1996*; *Comte, Guibaud & Baudu, 2006*). Therefore, proteins will be the basis of the synthetic EPS and the ratio of different chemicals will be based on the proteins. The compounds were mixed in ratios that were realistic (small enough concentration to be able to measure accurately using an analytical balance) to be acted as a corrosion inhibitor. They were mixed according to the following ratios:

a. Proteins:carbohydrates = 2.5:1

b. Proteins:humic substances = 6:1

c. Proteins:nucleic acids = 15:1

d. Proteins:uronic acids = 15:1

In order for the synthetic EPS to resemble the natural WAS EPS while keeping the complexity of the mixture low, one compound was selected from each chemical group.

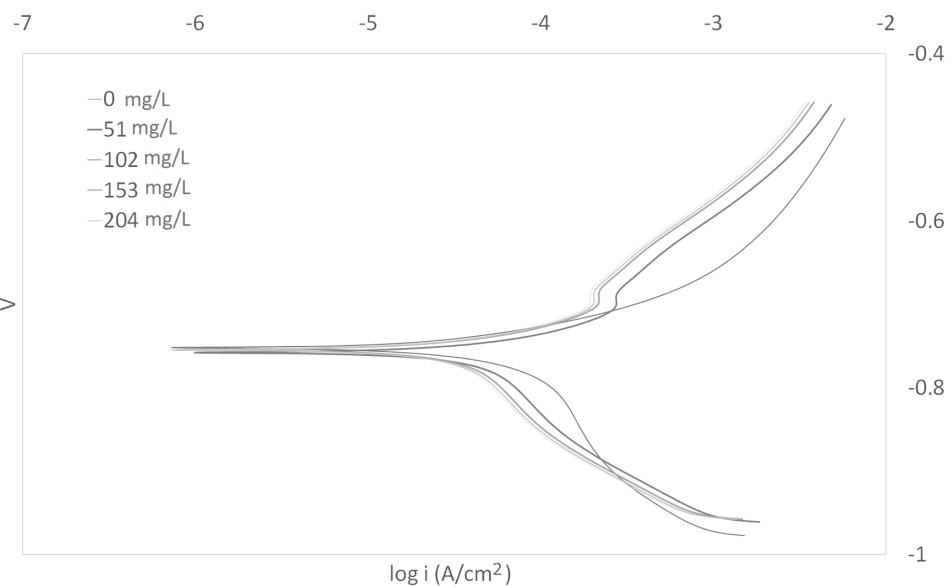

**Figure 2 Tafel plot for carbon steel in 3.64% NaCl concentrated with CO$_2$ with different concentrations of synthetic EPS at 50 °C.**

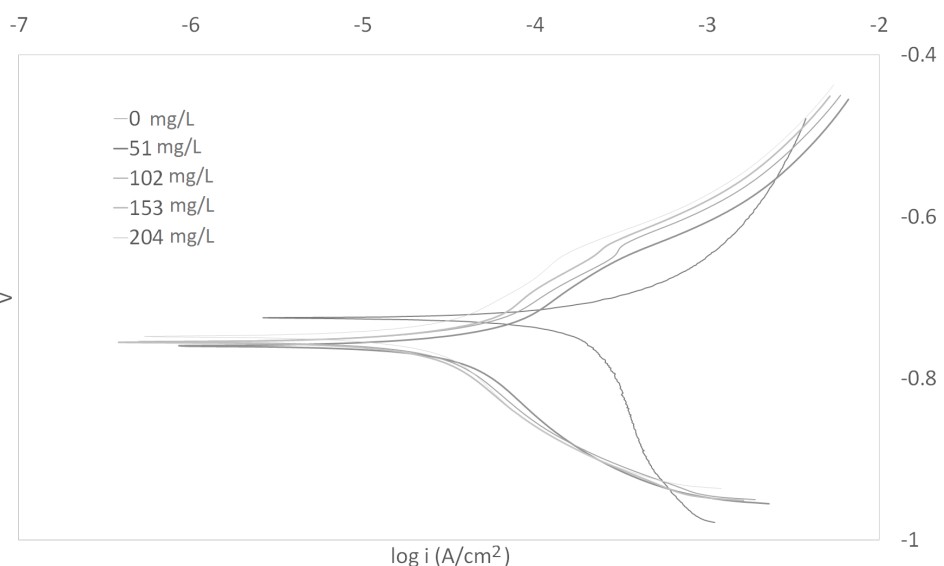

**Figure 3 Tafel plot for carbon steel in 3.64% NaCl concentrated with CO$_2$ with different concentrations of synthetic EPS at 70 °C.**

Since five major groups are generally being studied in the natural WAS EPS, five components were selected for the synthetic EPS formulation. They were chosen based on their structures and their chemical inhibition performances in the literature. Structure wise, compounds with nitrogen, oxygen, or sulfur atoms were preferred since all organic corrosion inhibitors typically contain at least one of these atoms, almost without exception. In addition, bigger compounds are also typically preferred as corrosion inhibitors because bigger compounds are more effective in separating the metal surface

**Table 2 Electrochemical parameters and the corresponding corrosion inhibition efficiencies in the presence of different concentrations of synthetic EPS at various temperatures.**

| Temperature (°C) | Temperature (K) | Concentration (mg/L) | $E_{corr}$ (V) | $I_{corr}$ ($\mu A/cm^2$) | Inhibition efficiency (%) | Surface coverage degree, θ |
|---|---|---|---|---|---|---|
| 25 | 298 | 0 | −0.73 | 52.48 | – | – |
| | | 51 | −0.71 | 25.12 | 68.72 | 0.6872 |
| | | 102 | −0.71 | 18.20 | 77.34 | 0.7734 |
| | | 153 | −0.71 | 14.45 | 82.00 | 0.8200 |
| | | 204 | −0.70 | 14.13 | 82.41 | 0.8241 |
| 50 | 323 | 0 | −0.74 | 125.89 | – | – |
| | | 51 | −0.76 | 37.15 | 86.03 | 0.8603 |
| | | 102 | −0.76 | 34.67 | 86.96 | 0.8696 |
| | | 153 | −0.76 | 33.11 | 87.55 | 0.8755 |
| | | 204 | −0.76 | 27.54 | 89.65 | 0.8965 |
| 70 | 343 | 0 | −0.73 | 223.87 | – | – |
| | | 51 | 0.76 | 45.71 | 91.70 | 0.9170 |
| | | 102 | −0.76 | 39.81 | 92.77 | 0.9277 |
| | | 153 | −0.76 | 34.67 | 93.71 | 0.9371 |
| | | 204 | 0.75 | 33.11 | 93.99 | 0.9399 |

**Table 3 Corrosion kinetic parameters for carbon steel in different concentrations of the synthetic EPS.**

| Inhibitor concentration (mg/L) | $E_a$ (kJ/mol) | $\Delta H_a^\circ$ (kJ/mol) | $\Delta S_a^\circ$ (J/mol K) |
|---|---|---|---|
| 0 | 27.46 | 24.81 | −71.28 |
| 51 | 12.54 | 8.75 | −131.24 |
| 102 | 15.21 | 12.56 | −120.83 |
| 153 | 17.24 | 14.59 | −115.63 |
| 204 | 16.47 | 13.82 | −118.73 |

from its corrosive environment when adsorbed on the metal surface. The compounds chosen for the synthetic EPS mixture fulfilled these descriptions, as illustrated in Fig. 9. Furthermore, those chemicals that had demonstrated corrosion inhibition were prioritized to be the candidates in the pool of selection. For protein, an amino acid, which is the building block of a protein was chosen. Glutamic acid, a common component of bacterial cell wall (*Hoare, 1963*), made an excellent candidate as an amino acid for the purpose of this study since it has also been proven to be an effective corrosion inhibitor in several studies (*Zhang et al., 2008a*, *2008b*). Glutamic acid showed approximately 54–90% of inhibition efficiency in 0.5M HCl with copper (*Zhang et al., 2008a*, *2008b*). Due to its potential in corrosion inhibition, it was chosen as the main component of the synthetic EPS. The second biggest composition was carbohydrate. For an organic corrosion inhibitor, typically, a bigger molecule is preferred. Carboxymethylcellulose (CMC), a relatively big molecular weight packed with multiple oxygen atoms, was selected as the candidate for the chemical group of carbohydrate. Its corrosion inhibition capability has

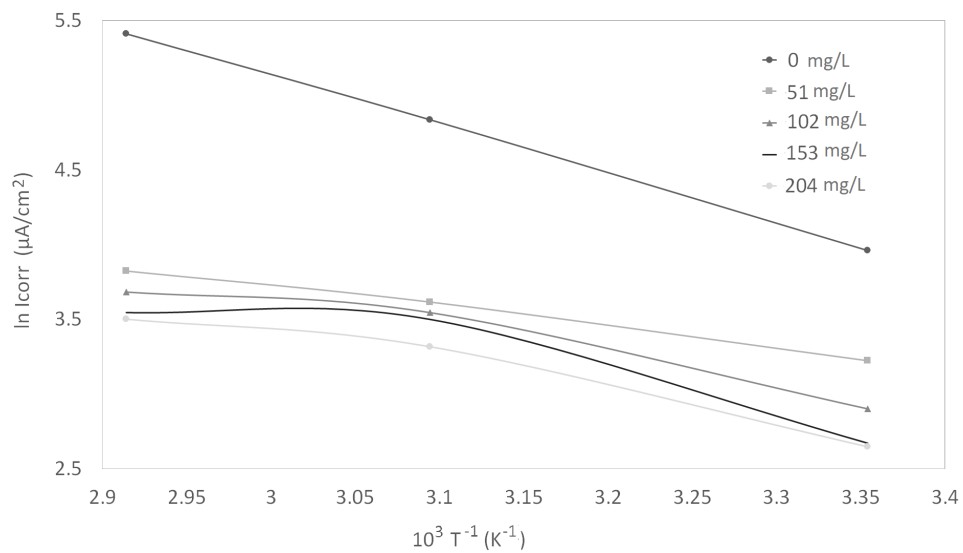

**Figure 4 Arrhenius plots of ln $I_{corr}$ vs $1/T$ at different concentrations of synthetic EPS.**

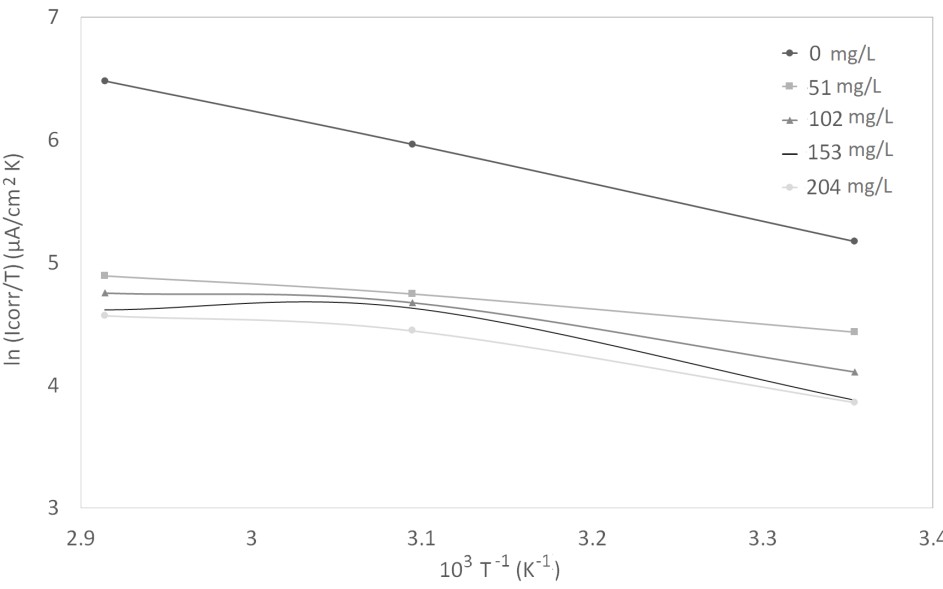

**Figure 5 Arrhenius plots of corrosion ln $(I_{corr}/T)$ vs $1/T$ at different concentrations of synthetic EPS.**

**Table 4 Thermodynamic parameters for the adsorption of synthetic EPS at different temperatures.**

| Temperature (°C) | Temperature (K) | $R^2$ | $K$ ((mg/L)$^{-1}$) | $\Delta G^{\circ}_{ads}$ (kJ/mol) | $\Delta H^{\circ}_{ads}$ (kJ/mol) | $\Delta S^{\circ}_{ads}$ (J/mol K) |
|---|---|---|---|---|---|---|
| 25 | 298 | 0.9980 | 0.01039 | −22.93 | 7.5192 | 102.11 |
| 50 | 323 | 0.9969 | 0.01325 | −23.53 | | 72.81 |
| 70 | 343 | 0.9999 | 0.01545 | −23.91 | | 69.68 |

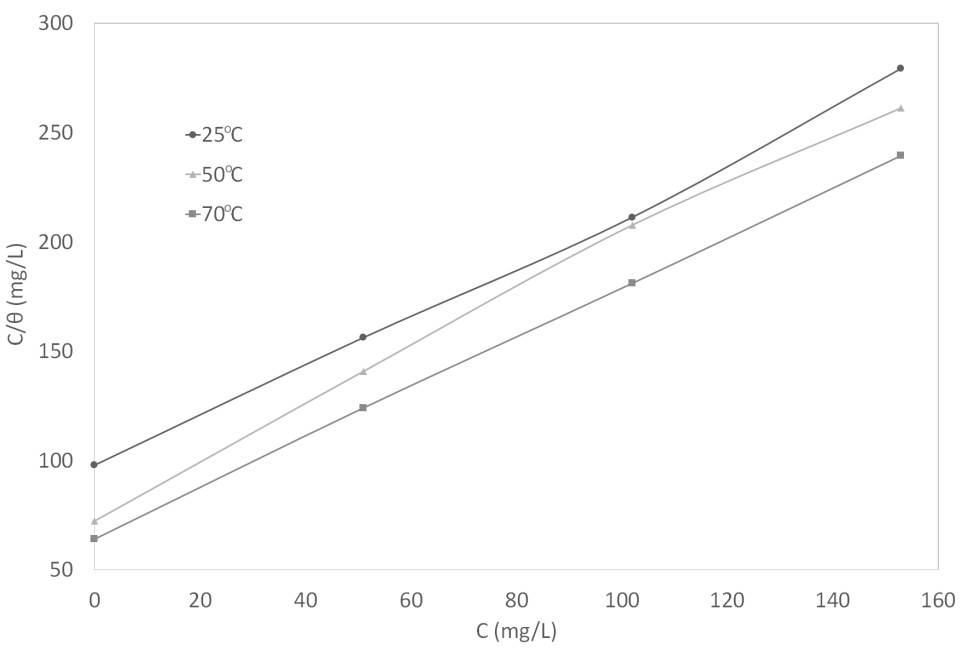

**Figure 6 Curve fitting of the experimental data to Langmuir isotherm.**

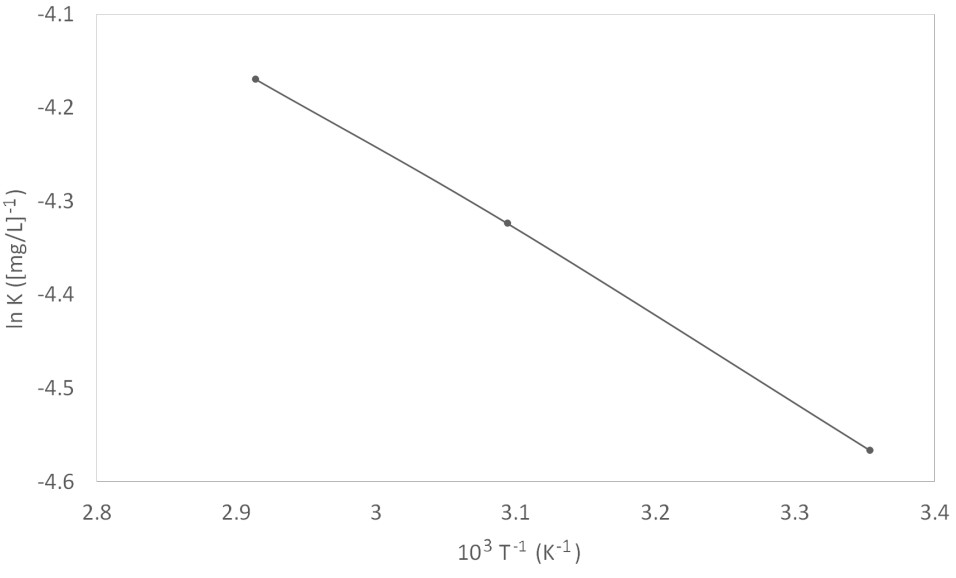

**Figure 7 Van't Hoff plot.**

also been proven excellent in various investigations (*Solomon et al., 2010*; *Bayol et al., 2008*). Inhibition efficiencies of about 65–72% were observed when CMC was used with mild steel in acid solutions $H_2SO_4$ (*Solomon et al., 2010*) and HCl (*Bayol et al., 2008*), respectively. However, corrosion inhibition studies on the rest of the chemical groups have no record in the literature to date. For humic substances and uronic acids, there are not many chemicals from these groups, so, humic acid and alginic acid were picked for

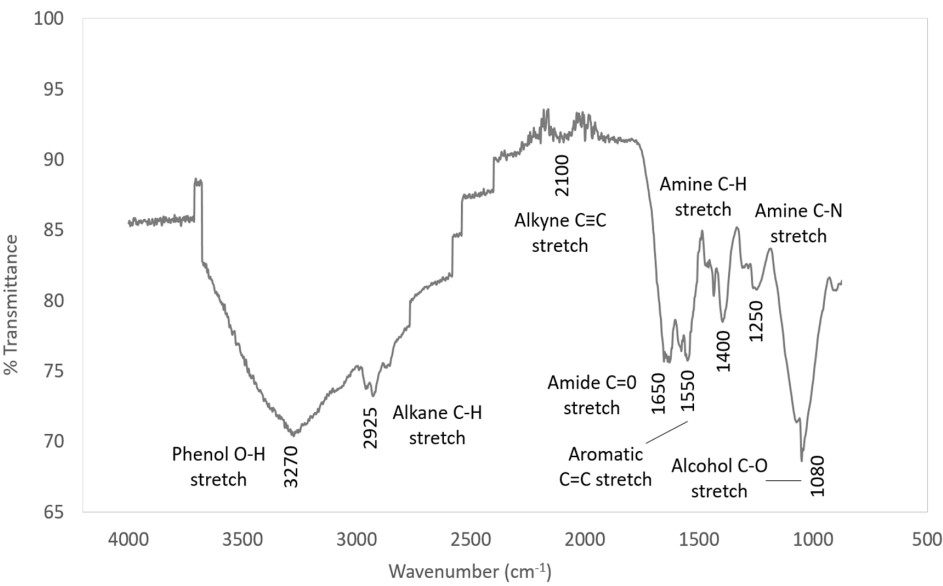

**Figure 8 IR spectra of synthetic EPS.**

**Table 5 Characteristic IR absorption frequencies of organic functional groups.**

| Characteristic absorptions ($cm^{-1}$) | Vibration type | Functional type |
|---|---|---|
| 3,200–3,600 | Phenol OH stretch | OH into polymeric compounds |
| 2,850–3,000 | Alkane C–H stretch | |
| 1,690–1,630 | Amide C=O stretch | Proteins |
| 1,590–1,650 | Amide (I) N–H bend | |
| 1,500–1,560 | Amide (II) N–H bend | |
| 1,350–1,480 | Alkane C–H bending | |
| 1,080–1,360 | Amine C–N stretch | |

each group, respectively. In the case of nucleic acids, there are only four choices in this group, namely thymine, guanine, adenine, and cytosine. Making a decision based on an economical point of view, the most affordable choice was thymine. Thymine is a relatively smaller compound compared to other chosen chemicals, but it contains both nitrogen and oxygen atoms, making it a desirable option. Hence, glutamic acid, CMC, humic acid, thymine, and alginic acid were chosen as the formulation for synthetic EPS. Their chemical structures are shown in Fig. 9.

The formulation of synthetic EPS was designed solely based on the chemical composition of natural WAS EPS, which was not optimized to meet the purpose of corrosion inhibition. Interestingly, this corrosion inhibitor showed corrosion inhibition performance comparable to the natural WAS EPS as well as commercial corrosion inhibitor. In order to improve the corrosion inhibition efficiency of synthetic EPS, the future direction of the current research will focus on optimizing the formulation to reduce the required applied concentration of corrosion inhibitor while achieving the maximum

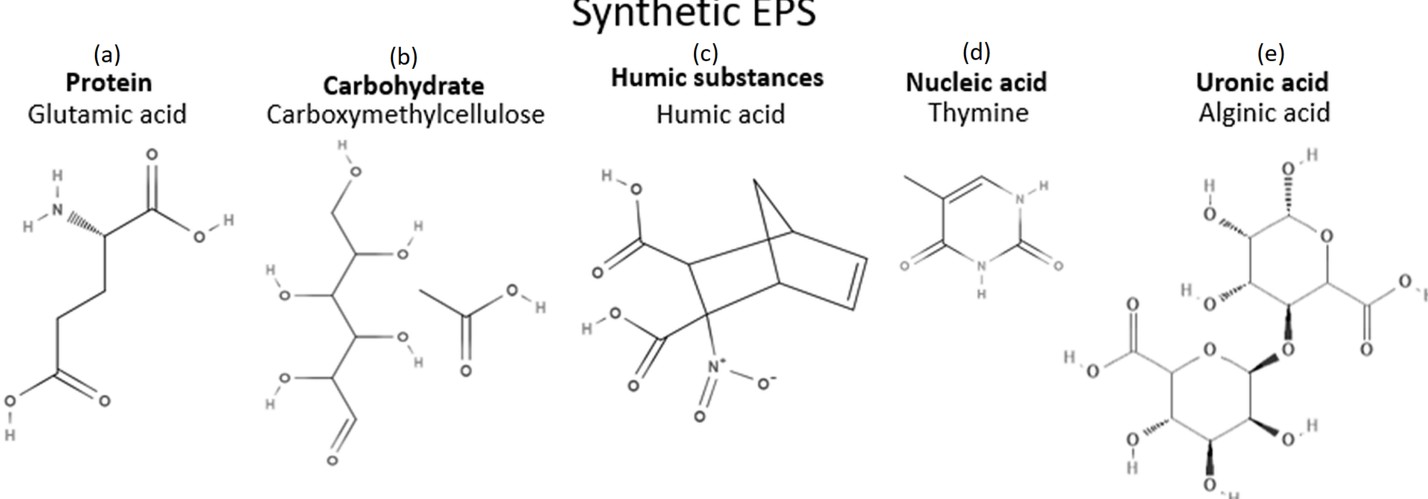

**Figure 9 Chemicals in synthetic EPS and their structures.** (A) Protein: Glutamic acid. (B) Carbohydrate: Carboxymethylcellulose. (C) Humic substances: Humic acid. (D) Nucleic acid: Thymine. (E) Uronic acid: Alginic acid.

attainable corrosion inhibition performance. This could be done by first reducing the number of compounds in the formulation, followed by optimizing the concentration of the compounds and the inhibition performance statistically.

The potential of utilizing biomass sources directly as corrosion inhibitors are undeniable. There is an enormous amount of studies on the application of plant extracts as corrosion inhibitors, but these products are still relatively rare in the market. One of the main reasons that is delaying the commercialization of these inhibitors could be the current immature resource recovery techniques. A lot of extraction methods are still economical infeasible these days. Therefore, in order to promote the use of renewable corrosion inhibitors, as well as to improve the marketability of these products, a bio-inspired corrosion inhibitor formulation is introduced in this study. Compared to the traditional plant extracts corrosion inhibitors, this type of renewable corrosion inhibitor is more market-ready because of several advantages: (1) renewable sources, (2) economic feasibility, (3) chemical composition consistency, as well as (4) corrosion inhibition performance consistency.

## Effect of concentration

The curves in Figs. 1–3 revealed well defined anodic and cathodic polarization Tafel regions. Note that only one set of experimental data was reported because the differences in triplicates were insignificant. The results for the triplicates can be found in the raw data section.

As observed in these figures, both cathodic and anodic reactions of carbon steel electrode corrosion were inhibited by the increase concentration of synthetic EPS in 3.64% NaCl saturated with $CO_2$ gas. This observation indicates that the addition of synthetic EPS reduced anodic dissolution as well as the hydrogen evolution reaction (*Morad, 1999*). This can be explained by the adsorption of inhibitor over the corroded surface

(*Abdallah, 2004*). Tafel lines of nearly equal slopes were obtained, indicating that the hydrogen evolution reaction was activated-controlled (*Bouklah et al., 2006*).

The details of electrochemical parameters obtained from the Tafel plots such as the values of corrosion potential, $E_{corr}$, corrosion current density, $I_{corr}$, corrosion protection efficiency, and surface coverage degree, θ, are presented in Table 2. The corrosion inhibition efficiency was calculated using Eq. (1), based on the $I_{corr}$ values, where $I_{corr,blank}$ and $I_{corr}$ were the corrosion current density without and with inhibitor, respectively. These values were obtained by the extrapolation of the cathodic and anodic Tafel lines to the corrosion potentials. The data showed that the $I_{corr}$ values decreased in the presence of synthetic EPS. These values also dropped as the concentration of inhibitor increased, meaning that the corrosion reaction was slowing down as the inhibitor concentration was increasing. This phenomenon can be attributed to the adsorption of synthetic EPS on the metal surface (*Bouklah et al., 2006*).

There was no definite pattern observed in $E_{corr}$ values in the presence of different concentrations of synthetic EPS. This result indicated that synthetic EPS may be considered as a mixed-type corrosion inhibitor (*Kumar & Yadav, 2016*) in the presence of $CO_2$ gas saturated 3.64% NaCl solution. The maximum displacement in $E_{corr}$ of less than 0.085 V suggests a mixed mode of inhibition (*Chaubey, Singh & Quraishi, 2015*). Mixed-type corrosion inhibitor retards corrosion rate by suppressing both anodic and cathodic corrosion reactions, typically by adsorbing on a metal surface, forming a protective film to reduce contact of metal surface from the corrosive environment (*Myrdal, 2010*).

The inhibition efficiency increased as the concentration of synthetic EPS increased. The maximum inhibition was about 94% with an optimum inhibitor concentration of 204 mg/L at 70 °C. At 25 °C, the maximum inhibition protection of synthetic EPS was 82% at a concentration of 153 mg/L. The previous study of WAS EPS inhibitor demonstrated an optimum inhibition performance of about 79% at a concentration of 1,000 mg/L (*Go et al., 2019*). Even though the inhibition performance showed only a mere improvement of 3%, the inhibitor concentration was reduced by about 6.5 times. It is known that the natural WAS EPS is rich in a variety of compounds. These compounds could have posed stearic hindrance on the adsorption of inhibition molecules on the metal surface, bring down the efficiency of the overall inhibition performance, so, higher concentrations of inhibitors were required to demonstrate the corrosion inhibition capability. Unlike the natural WAS EPS, the synthetic EPS was formulated specifically on the EPS groups that are known to perform as corrosion inhibitors. Hence, it is expected that the corrosion inhibition efficiency of synthetic EPS to be higher than the natural WAS EPS. Furthermore, in the case of commercial corrosion inhibitors, their corrosion protection performances are typically above 70%. Synthetic EPS has a corrosion inhibition performance that is within the range of commercial corrosion inhibitors. One advantage compared to natural WAS EPS is that its inhibition performance is consistent. The results obtained from this study strongly suggest the great potential commercialization value of synthetic EPS as a valuable material to inhibit corrosion issues in oilfield operations.

## Effect of temperature

The effect of temperature on the inhibited solution–metal reaction is highly complex because many changes could occur on the metal surface such as rapid etching and desorption of inhibitor, also, the inhibitor itself may undergo decomposition and/or rearrangement (*Bentiss, Lebrini & Lagrenée, 2005*). The effect of corrosion inhibition by synthetic EPS in NaCl solution saturated with $CO_2$ gas was studied with three different temperatures, that is, 25, 50, and 70 °C. Since the corrosion rate is greatly affected by the concentration of inhibitor as well as the temperature of the working environment, these factors have an important operational impact.

At different temperatures and inhibitor concentrations, the corrosion inhibition efficiencies varied. It was apparent that the rates of carbon steel corrosion, both in the blank solution of 3.64% NaCl saturated with $CO_2$ gas and with the presence of corrosion inhibitor, increased with increasing temperature. The impact of temperature on the overall corrosion reaction was more pronounced than the effect of inhibitor concentration. The inhibition efficiency increased with temperature. Typically, a decrease in inhibition efficiency with a rise in temperature suggests physisorption of the corrosion inhibitor. In contrast, an increase in inhibition efficiency with rise in temperature is indicative of a chemisorption mechanism (*Popova et al., 2003*). Therefore, the results clearly indicate a chemisorption mechanism of synthetic EPS on the carbon steel surface.

## Corrosion kinetic parameters

Corrosion kinetics parameters can be evaluated with different approaches. This study seeks to quantitatively evaluate the performance of the studied corrosion inhibitor (synthetic EPS) using an engineering calculation approach that yields empirical results. Instead of focusing mechanistically on the chemistry of the corrosion and corrosion inhibitor reactions to obtain the corrosion kinetics parameters, it is looking at an engineering perspective that is tailored to the studied system. The chemistry of the corrosion and corrosion inhibitor reactions are unutterably important, but it is not the center of the study. This study is not set up to investigate the mechanistic values of the corrosion kinetic and thermodynamic adsorption parameters.

This article emphasizes on the engineering significance of the synthetic EPS as a corrosion inhibitor by applying engineering equations that are based on basic corrosion theory but adapted heuristically to the studied system. Since the reported numbers (i.e., apparent activation corrosion energy, enthalpy of activation, entropy of activation) are empirical values that are only relevant to the studied system, these numbers may not be the duplicated with other system set up (e.g., traditional weight loss method with beaker testing). Even though the reported numbers are not the mechanistic values with respect to the theoretical corrosion reactions and the theoretical adsorption of the inhibitors, it serves a purpose to screen the performance of the studied corrosion inhibitor quantitatively using engineering calculations. In addition, the empirical values are also helpful in determining the behavior of the inhibitor in the studied system. For example, the enthalpy of activation could be used to describe whether the metal dissolution process is endothermic or exothermic. The idea is that, by adhering to the same experimental

set up and procedure, the same engineering calculations can be performed to estimate the same parameters for other inhibitors, serving as a comparison model. This is a particularly valuable heuristic tool, where the corrosion inhibitor can be screened rapidly. Thus, if the results obtained from the experiment are encouraging, further corrosion testing such as sparged beaker test and wheel test could be considered. Furthermore, the same approach has previously been demonstrated in the literature (*De Souza & Spinelli, 2009*; *Al-Amiery et al., 2014*), proven its usability and reliability.

The activation parameters for the corrosion reaction were calculated using an Arrhenius-type plot according to Eq. (2). It is worth mentioning that the Arrhenius equations applied were tailored to the studied system, as a heuristic approach to estimate the empirical values of the apparent activation corrosion energy, enthalpy of activation, and entropy of activation that are true to the system. $E_a$ in the equation denotes the apparent activation corrosion energy, $R$ is the universal gas constant, and $k$ is the Arrhenius pre-exponential factor. The values of apparent activation energy of corrosion were determined from the slope of $\ln I_{corr}$ vs $1/T$ plot, shown in Fig. 4. The data showed lower activation energy in the presence of inhibitors than in its absence, which is a typical pattern of chemisorption (*Bouklah et al., 2006*).

An alternative formulation of Arrhenius equation, that is, transition-state equation shown in Eq. (3), was used to calculate the change of enthalpy ($\Delta H_a^\circ$) and entropy ($\Delta S_a^\circ$) of activation for the activation complex formation in the transition state. In this equation, the $h$ is the Planck's constant, $N$ is the Avagadro's number, $\Delta S_a^\circ$ is the entropy of activation, and $\Delta H_a^\circ$ is the enthalpy of activation. Figure 5 shows a plot of $\ln (I_{corr}/T)$ against $1/T$ for synthetic EPS. A straight line was obtained with a slope of $\Delta H_a^\circ/R$ and an intercept of $\ln (R/Nh + \Delta S_a^\circ/R)$, from which the values of $\Delta H_a^\circ$ and $\Delta S_a^\circ$ were calculated. The positive enthalpy values reflected the endothermic nature of metal dissolution process. Large and negative values of entropy imply that the activated complex in the rate determining step represents an association rather than a dissociation step (*Bouklah et al., 2006*).

$$I_{corr} = ke^{-\frac{E_a}{RT}} \tag{2}$$

$$I_{corr} = \frac{RT}{Nh} \exp\left(\frac{\Delta S_a^\circ}{R}\right) \exp\left(-\frac{\Delta H_a^\circ}{RT}\right) \tag{3}$$

## Thermodynamic adsorption parameters

Adsorption isotherms provide insights into the interaction among the adsorbed molecules and the metal surface, which can help to better understand the corrosion inhibition mechanism. Similar to the corrosion kinetic parameters, the thermodynamic adsorption parameters reported in this section are only empirical values relevant to this studied system. The values of surface coverage ($\theta$) to different concentrations of inhibitor, obtained from the polarization measurements in the temperature range of 25 to 70 °C (298–343 K) were used to explain the best isotherm to determine the adsorption mechanism.

The values of θ were assumed to be the corrosion inhibition efficiencies. The reason being, without the presence of inhibitor compound, an inhibition efficiency of 0% is expected, so, when an inhibitor compound is introduced to a corrosive environment, the improved corrosion inhibition efficiency is believed to be solely contributed by the coverage of the inhibitor compound on the metal surface. The surface coverage, θ, were used in a series of equations shown in Eqs. (4–6) (*Khamis, 1990*). Equation 4 showed the relationship of $I_{corr}$, $I_{corr,blank}$, $I_{sat}$, and θ. $I_{sat}$ is the current density of entirely covered surface. This equation was then be rearranged into Eq. (5). As $I_{corr}$ was greater than $I_{sat}$, Eq. (5) was simplified to Eq. (6).

$$I_{corr} = (1 - \theta)I_{corr,blank} + \theta I_{sat} \tag{4}$$

$$\theta = \frac{I_{corr,\,blank} - I_{corr}}{I_{corr,blank} - I_{sat}} \tag{5}$$

$$\theta = \frac{I_{corr,\,blank} - I_{corr}}{I_{corr,blank}} \tag{6}$$

In the range of temperature and inhibitor concentration studied, the best correlation between the experimental results and the adsorption isotherm functions was obtained using Langmuir adsorption isotherm. The Langmuir isotherm for monolayer adsorption is given by Eq. (7). By linearizing this equation, Eq. (8) was obtained.

$$\frac{\theta}{1 - \theta} = KC \tag{7}$$

$$\frac{C}{\theta} = \frac{1}{K} + C \tag{8}$$

In Eqs. (7) and (8), θ is the surface coverage degree, $C$ is the inhibitor concentration in the NaCl solution, and $K$ is the equilibrium constant of the adsorption process. The correlation coefficient, $R^2$, was used to describe how close the isotherm fits the experimental data. The plot of $C/\theta$ against $C$ gave a straight line and the linear correlation coefficients were fairly close to 1, indicating good fit to the data. This graph is shown in Fig. 6. The adsorption behavior of synthetic EPS conformed to Langmuir isotherm, suggesting monolayer adsorption, which is a typical behavior of chemisorption (*Christmann, 2012*).

In general, Langmuir isotherm is not recommended to be used to describe a mixture system because the individual components in a mixture can each be adsorbed in different ways. However, it is applicable to this study because the synthetic EPS, as a corrosion inhibitor formulation, was treated as an entity. The individual contribution of compounds in the mixture of synthetic EPS were considered unimportant, therefore, being omitted. These are the basic assumptions of Langmuir isotherm: (1) surface of the adsorbent (metal) is uniform, (2) adsorption sites are equivalent, (3) adsorbed molecules do not interact, (4) all adsorption occurs through the same mechanism. Assuming that the metal

surface is uniform, the adsorption sites are equivalent, and the inhibitor formulation is being treated as an entity, Langmuir isotherm is appropriate to be used to describe the overall adsorption mechanism. There are numerous studies in the literature where Langmuir was used to describe the adsorption mechanism of an inhibitor mixture, especially plant extracts (*Ebenso, Eddy & Odiongenyi, 2008*; *Ating et al., 2010*).

$K$ values were calculated from the intercepts of the same plot (Fig. 6). The constant of adsorption, $K$, can be related to the standard free energy of adsorption, $\Delta G^{\circ}_{ads}$, using Eq. (9). The constant $1 \times 10^6$ in the equation is the concentration of water molecules expressed in mg/L, $R$ is the universal gas constant, $T$ is the absolute temperature. On the other hand, $\Delta H^{\circ}_{ads}$ can be deduced from the integrated version of the Van't Hoff equation expressed by Eq. (10). Figure 7 shows the plot of ln $K$ vs $1/T$ which yield a straight line with a slope of $-\Delta H^{\circ}_{ads}/R$. The value obtained was used to find the $\Delta H^{\circ}_{ads}$. The calculated $\Delta H^{\circ}_{ads}$ was then used to calculate the values of $\Delta S^{\circ}_{ads}$ by using Eq. (11).

$$\Delta G^{\circ}_{ads} = -RT \ln\left(1 \times 10^6 \ K\right) \tag{9}$$

$$\ln K = -\frac{\Delta H^{\circ}_{ads}}{RT} + \frac{\Delta S^{\circ}_{ads}}{R} \tag{10}$$

$$\Delta G^{\circ}_{ads} = \Delta H^{\circ}_{ads} - T\Delta S^{\circ}_{ads} \tag{11}$$

A more in-depth study of the inhibitor adsorption mechanism was investigated using the values of thermodynamic parameters. The details can be found in Table 4. The spontaneity of the adsorption of inhibitor on the metal surface as well as the stability of the adsorbed layer on the metal surface was demonstrated by the resulted negative values of $\Delta G^{\circ}_{ads}$. Typically, an endothermic adsorption process that has a positive value of $\Delta H^{\circ}_{ads}$ is attributed unequivocally to chemisorption, while an exothermic adsorption process with $\Delta H^{\circ}_{ads}$ of negative value may involve either physisorption or chemisorption, or a combination of both the processes (*Bentiss, Lebrini & Lagrenée, 2005*). In this study, the $\Delta H^{\circ}_{ads}$ was positive, once again implying a chemisorption mechanism. The value of $\Delta S^{\circ}_{ads}$ decreased with increased temperature, implying that the reaction of adsorption was getting less disordered.

### Fourier-transform infrared spectroscopy

The corrosion inhibition capability of synthetic EPS was proven significant in this study. The trend in the IR spectrum of the synthetic EPS followed closely to the natural WAS EPS (*Go et al., 2019*) as expected because it is formulated based on the chemical composition of natural WAS EPS. Similar to the natural WAS EPS, the FTIR results of synthetic EPS showed that functional groups O–H, N–H, C–N, C=O, and C–H were present. Since the synthetic EPS and natural WAS EPS both have the same functional group, it can be deduced that these functional groups play major roles in corrosion inhibition. Other authors have also suggested the contribution of these

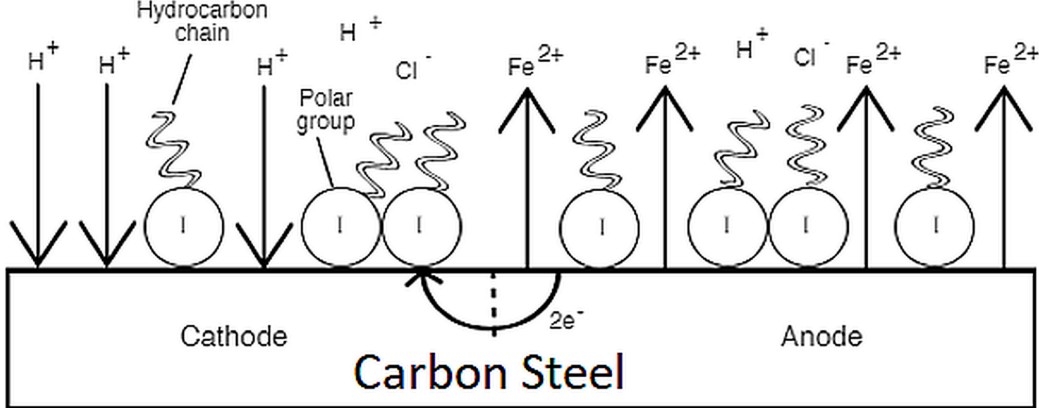

**Figure 10 Corrosion inhibition mechanism of synthetic EPS.**

functional groups in corrosion inhibition (*Ebenso, Eddy & Odiongenyi, 2008*; *Ating et al., 2010*).

This study treated the mixture of synthetic EPS as an entity, meaning that the contribution to the overall inhibition cannot be ascribed to any single component in the mixture. A suggestive corrosion inhibition mechanism of the synthetic EPS can be explained by the electrochemical theory. The electrochemical theory of corrosion holds that the metal surface corroding in an electrolyte is covered with local electrolytic cells. Some areas of the metal can act as anodes and other areas can act as cathodes, shown in Fig. 10, depending upon the history of the metal regarding heat treatment, presence of imperfections, scratches, greases, paint coatings, fingerprint smudges, etc. At anodic sites, the metal usually dissolves into solution. Electrons given from these sites are transported to local cathodes and collected by electron acceptors such as hydrogen ions and oxygen. As previously suggested, synthetic EPS acts as a mixed-type corrosion inhibitor, meaning that the molecules in the synthetic EPS chemisorbed on both the anodic and cathodic sites of metal surface to form a monolayer protection film. The functional groups rich in nitrogen and oxygen atoms acted as the polar head of organic corrosion inhibitors, adsorbing on metal surface by forming chemical bonds between the inhibitor molecules and metal ions, while the non-polar hydrocarbon chain attached to the polar head isolated the metal surface from the corrosive surrounding, suppressing both anodic and cathodic corrosion reactions, reducing the overall corrosion rate.

## Engineering, operation, and economic significance of synthetic EPS

The corrosion inhibition performance of the synthetic EPS shown in this study is promising. Other areas that are interesting to be investigated are the engineering, operational, and economical sides of this corrosion inhibitor formulation.

This article is an extension of a previous study that showed the potential of EPS extracted from WAS of wastewater treatment operations as a green corrosion inhibitor for

$CO_2$ corrosion. A maximum inhibition performance of about 80% was achieved with the application of 1,000 mg/L of this inhibitor (*Go et al., 2019*; *Go, Holmes & Hernandez, 2019*). The major flaw in this corrosion inhibitor is that the inhibition performance is not consistent due to the variation in wastewater compositions. Thus, synthetic EPS was formulated in this study based on the natural composition of WAS EPS, as a bio-inspired material, that resemble the inhibition performance of natural WAS EPS, but with consistent inhibition performance. This approach is novel in research and development of green corrosion inhibitor development. Commercial corrosion inhibitor for the oil and gas industry are generally petroleum-based, while most green corrosion inhibitors reported in the literature were directly extracted from agricultural sources. The technique used in this study for the development of bio-inspired material as a corrosion inhibitor can be further expanded in the area of green corrosion inhibitors development and potentially be extended to other research fields.

In terms of operation significance, similar to most bio-inspired materials, this corrosion inhibitor formulation can be formulated with commercial renewable resources or extracted from natural resources (WAS), leaving a lesser environmental impact compared to the commercial petroleum-based corrosion inhibitors. Besides having corrosion inhibition performance comparable to commercial products (commercial products usually show inhibition performance above 70%), the synthetic EPS is also just as easy to be applied like commercial corrosion inhibitors, making it an excellent alternative.

The economic analysis of synthetic EPS was evaluated in this study. The production cost of the synthetic EPS is about $4.23 for every 10,000-inhibition treatment (assuming one L system/treatment), while the market price of a typical commercial oil and gas corrosion inhibitor costs about $2.38 per 10,000 applications. It is worth mentioned that the synthetic EPS is formulated based solely on the composition of natural EPS. The economic feasibility can be improved in future studies by product optimization to reduce the applied inhibitor concentration and enhance the inhibition performance.

It is evident that bio-inspired systems/materials have high potential in revolutionizing the current market to reduce dependence on fossil fuel-based products as well as to promote innovative product development approach. This transformation is not only applicable in the corrosion inhibitor industry but should also be extended to benefit other research and development areas.

## CONCLUSIONS

The studied corrosion inhibitor, synthetic EPS, showed corrosion inhibition capability for carbon steel when tested in 3.64% NaCl saturated with $CO_2$ gas. Synthetic EPS is a surrogate of biomass-based corrosion inhibitor inspired by sources with varied chemical compositions to overcome the composition inconsistency in biomass that can cause unreliable corrosion inhibition performance. Synthetic EPS is a mixture of glutamic acid, CMC, humic acid, thymine, and alginic acid, following the chemical composition of natural WAS EPS extracted by heating method. Unlike the natural WAS EPS that is rich in assorted of molecules that could promote stearic hindrance on the adsorption of

inhibitor molecules, synthetic EPS was designed specifically based on the EPS groups that are known to perform as corrosion inhibitors. The electrochemical testing used in this study showed that the corrosion rates were significantly reduced with the addition of synthetic EPS. With concentration of 204 mg/L in 3.64% NaCl saturated with $CO_2$ gas, synthetic EPS showed maximum corrosion inhibitions of 82.41%, 89.65%, and 93.99% at 25, 50, and 70 °C, respectively. Its performance compared favorably with natural WAS EPS and commercial corrosion inhibitors. It was found that the inhibition performance was controlled by both the concentration of inhibitor and temperature.

The corrosion inhibition capability was due to chemisorption shown by several evidences:

1. An increase in corrosion inhibition efficiency with increase temperature;
2. A decrease in activation energy in the presence of inhibitor;
3. Endothermic adsorption.

Since the formulation of synthetic EPS was designed solely based on the chemical composition of natural WAS EPS, it was not optimized to meet the purpose of corrosion inhibition. Based on the results presented and the needs and requirements of corrosion protection service providers, the future direction of the current research will focus on optimizing the formulation in order to reduce the required applied concentration of corrosion inhibitor while achieving the maximum attainable corrosion inhibition performance. This could be done by first reducing the number of compounds in the formulation, then optimizing the concentration of the compounds and the inhibition performance statistically.

This bio-inspired material is developed in the hope to promote the commercialization of renewable corrosion inhibitors. According to the Google Scholar database, there are as many as 17,600 publications on the topic of green corrosion inhibitors from the year 1980 to 2018. A significant portion of these publications were made up by phytochemical-based compounds. Plant extracts are gaining popularity as green corrosion inhibitors candidates not only because they are renewable sources, but also their potential in corrosion mitigation. On average, the attainable corrosion protection efficiencies of these inhibitors can range from 70% to as high as 98% (*Eddy & Ebenso, 2008*; *Ibrahim, Chehade & Zour, 2011*; *Elmsellem et al., 2014*; *da Rocha, Gomes & D'Elia, 2014*; *Sangeetha et al., 2012*). Despite the overwhelming research evidence that suggests the impressive performance of these inhibitors, there are still drastic uneven numbers of research reports and products on the shelves. Needless to say, resource recovery efforts take time to improve. Before these techniques are optimized, or if there are enough products to be added to the product line to improve the cost issue (*Go et al., 2019*), until then, bio-inspired material could be an alternative to balance the environmental problems bring by petroleum-based corrosion inhibitors and the economic complications raise by plant extracts corrosion inhibitors. As a matter of course, the idea of benefiting from bio-inspired systems and materials will not only benefit the corrosion inhibitor sector but is prompt to be extended to any other applicable research area.

### Funding
This work was supported by the Coastal Chemical Research Fund (Project No.: 14-1227). The funders had no role in study design, data collection and analysis, decision to publish, or preparation of the manuscript.

### Grant Disclosures
The following grant information was disclosed by the authors:
Coastal Chemical Research Fund: 14-1227.

### Competing Interests
Tre Bertrand is employed by Coastal Chemical Co. LLC.

### Author Contributions
- Liew Chien Go conceived and designed the experiments, performed the experiments, analyzed the data, performed the computation work, prepared figures and/or tables, authored or reviewed drafts of the paper, and approved the final draft.
- Dilip Depan conceived and designed the experiments, authored or reviewed drafts of the paper, and approved the final draft.
- William E. Holmes conceived and designed the experiments, authored or reviewed drafts of the paper, and approved the final draft.
- August Gallo conceived and designed the experiments, authored or reviewed drafts of the paper, and approved the final draft.
- Kathleen Knierim conceived and designed the experiments, authored or reviewed drafts of the paper, and approved the final draft.
- Tre Bertrand conceived and designed the experiments, authored or reviewed drafts of the paper, and approved the final draft.
- Rafael Hernandez conceived and designed the experiments, authored or reviewed drafts of the paper, and approved the final draft.

### Data Availability
The raw measurements are available in the Supplemental File.

### Supplemental Information
Supplemental information for this article can be found online at http://dx.doi.org/10.7717/peerj-matsci.4#supplemental-information.

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
