# Peer review of "Kinetic and thermodynamic analyses of the corrosion inhibition of synthetic extracellular polymeric substances"

_PeerJ Materials Science, doi:10.7717/peerj-matsci.4_

## Round 0.1 · original submission · Major Revisions

Both reviewers raise concerns that have to be addressed. I may suggest to follow the suggestions of Reviewer 1 and to perhaps conduct standard corrosion tests instead/in addition to Tafel plots. In order Tafel plots to be publishable, you may consider to address the comments 1- 5 of Reviewer 1 and to address the experimental concerns of Reviewer 2 (old vs new sample for every run, etc)

·

Basic reporting

Language is satisfactory, when all other basic issues are completely unsatisfactory.

(1) The authors misunderstand the origin of Equations which they apply to treat the data.

(2) The authors' background in kinetics of corrosion processes is very poor.

(3) Presentation of electrochemical data and corresponding quantities is misleading.

(4) There is no any scientific result or hypothesis. Instead, we see a number of observations for very complex and less characterized system accompanied by incorrect data treatment, see below.

Experimental design

The arrangement of experiment is more or less traditional for corrosion research, based mostly on the measurements of polarization curves. Very important information about the type of carbon steel is absent (the authors report only elemental composition, but do not consider phase composition which is crucial for corrosion behavior) and escape to describe the samples surface pretreatment (which is crucial for kinetics of interfacial processes).

The most dramatic problem is the choice of the inhibitor under study. For practical applications, this mixture can probably work, but it is impossible to consider the mixture of 5 substances in terms of adsorption isotherms. Any of the components is characterized by certain isotherm when adsorbs in the absence of other species. However the adsorption from the mixture is typically competitive, and cannot be considered in any additive manner. The parameter of Langmuir adsorption isotherm has a meaning of surface-adsorbate interaction energy and cannot be extracted from the data for mixtures.

For FTIR experiments, I failed to understand under what conditions these spectra were registered. To consider FTIR information in the context of adsorption from solution, one needs to measure the spectra in situ. The usual problem in this case is the overlap of bands for solute and adsorbate, and there are many specific techniques to separate these contributions, but nothing is written about this principle technical issue. Most probably Fig.8 demonstrate the spectra of the mixture as is, so it can tell us nothing about participation of certain functional groups in adsorption.

There is no indication of solution pH, which is important in respect to protonation constants for acids. The adsorption properties of organic acids and their anions are completely different.

Validity of the findings

All the findings resulting from the treatment of polarization curves are senseless because of the following reasons.

(0) There are no linear parts in Tafel plots, so I wonder how the authors extrapolated the anodic and anodic curves in their Tafel plots to obtain I_corr. I can assume that one of the reasons of non-linear behavior is the Ohmic drop because the authors use a huge (5cm^2) electrode, and the currents in rather wide potential intervals are above 1 mA. For any electrochemical experiments, much smaller electrodes must be applied, and Ohmic contribution should be examined from the very beginning.

(1) Arrhenius Eq (2) is valid for the rates of particular processes, and cannot be applied to I_corr, which presents combination of at least two (anodic and cathodic) coupled processes. It is very easy to see (Figs 1-3) that the inhibitor effects on cathodic and anodic branches are different, so the contribution in I_corr are also different for solutions of various concentrations.

(2) It is impossible to determine the surface coverage from I_corr using Eq. (4) or even from the currents at cathodic and anodic branches because the rates of reactions are not obligatory proportional to adsorbate-free surface. In particular, anodic dissolution includes chloride adsorption step, which can preferentially occur at some active centers. Adsorption of large molecules like carbohydrates does not obligatory suppress the adsorption of low-molecular species participating in corrosion. For hydrogen evolution, the rate can be proportional either to teta, or to teta^2, depending on the nature of the limiting step.

(3) It is impossible to apply Langmuir isotherm to high surface coverages (0.69 an even much higher) because this isotherm assumes the absence of lateral interactions in adlayer. These interactions are unavoidable for molecules with any functional groups. It is also impossible to apply Langmuir isotherm to large molecules which tend to poly-layer adsorption. Moreover, before starting with any isotherm one needs to clarify whether the adsorption is reversible or irreversible. Correspondingly, 'adsorption part' is completely mistaken.

(4) I can assume that 'local electrolytic cells' mentioned by the authors is the notion related to local galvanic pairs. However this mechanism of corrosion is not universal, it can take place only for strongly inhomogeneous surface and depends on distribution of minor components in the steel sample. In any case, in NaCl solution the basic mechanism is related to chloride, not to any galvanic pairs.

(5) The data tabulated by the authors do not allow to determine any quantities with the accuracy presented now (up to five significant digits). What is tabulated for K (three points in Table 4) does not allow to calculate anything even with low accuracy.

Additional comments

If the authors consider their mixture as useful for the practical corrosion protection, they should arrange standard corrosion tests instead of measuring polarization curves.

If the authors plan to develop corrosion science and to publish articles, they should start from the very beginning.

Reviewer 2 ·

Basic reporting

In this work English is suitable, clear and technically correct.
However the introduction is not enough described and does not clearly introduce the innovative aspects of the study and the literature can be more descriptive on the actual background. (The authors shoud be careful when citing the literature with a specific program since sometimes the references can be missing)
The structure of the article is not good in the results part where no description is given and should be more emphasized, maybe with a joint results and discussion section.
The number of figures is adequate for the present study and are relevant for the clarity of the paper, however they can be more thoroughly described throughout the text helping the reader to follow the line of the work. In particular figure9 is not described in the text
The results could be relevant to the work if more elaborated

Experimental design

The present study is within the scope of the journal, could be of a particular scientific interest and is quite innovative for the use of synthetic EPS as corrosion inhibitors.
The aim of the work should be more emphasized in terms of applicability of the inhibitor in the introduction section.
The experimental part could be better defined. The authors should better clarify the methods used for statistical reproducibility and repeatability. If the authors used the same sample and performed replicated electrochemical analyses, it is possible that the results obtained where not comparable to each other since the analyses induced a change from the original electrochemical conditions and an anomalous variability. If the authors used different samples, they should explicit the number of replicates more clearly in order to have a good data dispersion, a statistical validation and a significance of the measurements. Why did the authors choose a range of temperatures from 25 °C to 70°C? Can the authors report it in the experimental section to justify the choice of temperature range?
Do the authors think that 6 hours is enough to define the behavior of the systems in presence of inhibitors? It would be important to justify it. Did the authors evaluate the Fe release inside the solution to verify the inhibition effect of the polymer?

Validity of the findings

The impact and the novelty of the work is undeniable but the work lacks some rationale in the methodology used. For instance in the section where the authors discussed the properties of synthetic EPS is a good introduction part but is not supported by the results previously described, thus should be moved or both in the introduction and in the methodology section.
The data provided in the text can be questionable if the Tafel experiments were performed always on the same sample. The analyses changed the interface and the behavior described does not reproduce the actual conditions, inducing an anomalous variability. It would be better perform experiments using for each condition a 'new' sample (3 samples for each condition).
Some parts of the results are missing and are described in the discussion section. I would suggest to join the results and discussion section in order to have a text that results more clear to the reader.

Additional comments

In this work the author investigated the role of specific synthetic EPS to be evaluated as corrosion inhibitors for mild steel in an anaerobic chloride-rich environment. The present study is within the scope of the journal, could be of a particular scientific interest and is quite innovative for the use of synthetic complex polymers that can be found in nature. However the work needs more thorough investigations and is not enough elaborated to be published as a scientific article.
I suggest the authors to rewrite the article focusing on the results and discussion section, clearly indicating the results ad describing the trends found and after that trying to justify the data obtained according to the parameters chosen.
In terms of scientific rigor, the units should always be the same throughout the paper (use K or °C and not both)

---

## Round 0.2 · Minor Revisions

Some additions and changes were made but they are not sufficient to make the manuscript publishable as is. I may suggest to downplay the thermodynamic and kinetic analysis, as pure empirical data; remove mentioning of Langmuir isotherm and 'monolayer coverage' from the abstract and conclusions. It would help to simplify the message (e.g. the formulation as tested provided a significant decrease in corrosion rate as compared to control'). Also, provide some more rational on how this complex blend of 5 components was arrived to. Was it optimized? Was it based on a natural composition of some kind? Is the observed reduction of corrosion practically important and significant? Any insights on which component of the blend is responsible for the observed effect? Note that the components cannot be treated as a single component, as their chemistries are quite different, and the polymer absorption is irreversible (which is contrary to the assumption of Langmuir isotherm).

---

## Round 0.3 · accepted · Accept

The comments were addressed and the paper can now be published